# Knowledge-Consistent Dialogue Generation with Knowledge Graphs

## Abstract

Pre-trained generative language models have achieved impressive performances on dialogue generation tasks. However, when generating responses for a conversation that requires complicated factual knowledge, they are far from perfect, due to the lack of mechanisms to retrieve, encode, and reflect the knowledge in the generated responses. Unlike the methods working with unstructured text that are inefficient in retrieving and encoding the knowledge, some of the knowledge-grounded dialogue generation methods tackle this problem by leveraging the structured knowledge from the Knowledge Graphs (KGs). However, existing methods do not guarantee that the language model utilizes a relevant piece of knowledge for the given dialogue, and that the model generates dialogues which are consistent with the knowledge, from the KG. To overcome this limitation, we propose **SU**bgraph **R**etrieval-augmented **GE**neration (**SURGE**), a framework for generating knowledge-consistent, context-relevant dialogues with a KG. Specifically, our method first retrieves the relevant subgraph from the given KG, and then enforces consistency across the facts by perturbing their word embeddings conditioned on the retrieved subgraph. Then, it learns the latent representation space using graph-text multi-modal contrastive learning which ensures that the generated texts have high similarity to the retrieved subgraphs. We validate the performance of our SURGE framework on the OpendialKG dataset and show that our method does generate high-quality dialogues that faithfully reflect the knowledge from the KG.

## 1 Introduction

Dialogue systems aim at conversing with humans by generating human-like responses, considering the context and history of the dialogue. Recently, thanks to the development of pre-trained language models (PLMs) for text generation [32, 34], neural dialogue agents are able to generate fluent responses. However, despite their satisfactory fluency, they often generate factually incorrect responses due to a lack of explicit knowledge. The problem can become worse, when the conversation requires accurate knowledge about certain subjects. Thus, to overcome such limitations, some of the recent methods access the external knowledge sources, for example, Wikipedia [7] or Web [21], and then retrieve the documents containing the relevant knowledge for ongoing conversations.

While retrieving the relevant documents from a large-scale text corpus with information retrieval techniques significantly boosts the performance of dialogue agents [18, 24], the computational burden of searching for the relevant documents and embedding them on the fly could be high, which may compromise the responsiveness of the conversation agent. Thus, we instead consider the approach that utilizes the pre-compiled Knowledge Graph (KG) [2, 43] consisting of symbolic facts, which represent the entities as nodes and their relations as edges, in the form of a triplet, e.g., *(Pride & Prejudice, written by, Jane Austen)*. Such KG-augmented dialogue generation models are highly efficient compared to retrieving from and augmenting with unstructured texts. This is because we

Submitted to 36th Conference on Neural Information Processing Systems (NeurIPS 2022). Do not distribute.

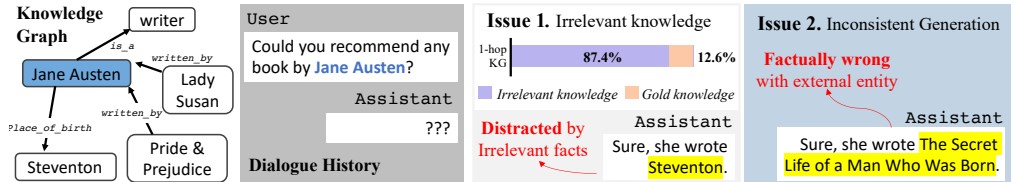

Figure 1: **Motivation.** Existing knowledge-grounded dialogue generation models with KG utilize the multi-hop subgraph for entities in the dialogue context (Jane Austen). However, they suffer from the following two problems: **(1) irrelevant knowledge** where only 12.6% of facts from 1-hop KG are useful to generate the target responses given a dialogue context, and **(2) inconsistent generation** including the factually wrong statement.

can directly retrieve entities from the context without searching for all candidate documents from a large text corpus (e.g. Wikipedia), and the retrieved facts succinctly encode the required knowledge in the most compact and effective form. Figure 1 shows an example which illustrates how the KG represents facts as relations among entities, that help generate a knowledge-grounded response.

Few recent works [41, 9, 50] use the KG to provide facts associated with the entities in the dialogue context to the conversation agents. However, they utilize all the triplets associated to the given entity, whose facts are mostly irrelevant to the dialogue context (e.g., Jane Austen was born in Steventon in Figure 1), which could mislead the model into generating factually incorrect responses. We found that about 87% of facts from 1-hop KG are irrelevant to the context in the OpendialKG dataset [29]. Moreover, encoding all the facts including the unnecessary ones is computationally inefficient [9].

Even after correctly retrieving the relevant facts from the KG, it is not straightforward to combine the representations from two heterogeneous modalities: the dialogue context is represented as a text, meanwhile, the knowledge is represented as a graph. Moreover, since the PLMs already have tons of pre-trained parameters trained on the unstructured texts, properly conditioning the structured graph to the PLM is highly important. If not done so, PLMs may generate inconsistent responses with regard to the knowledge from the retrieved subgraph, whose phenomenon is known as hallucination [37, 8] where PLMs generate responses with their own memorized yet incorrect knowledge.

In this work, we tackle such challenging and fundamental issues of knowledge-consistent dialogue generation with KG[1]. In particular, at the first step, we propose a context-relevant subgraph retrieval that retrieves only the relevant triplets from a large KG to prevent the model from generating context-irrelevant responses. Notably, our subgraph retrieval method is end-to-end trainable jointly with the generation objective by marginalizing the likelihood of the generated sentences over the latent variable of the retrieved subgraph [24]. Then, to encode the retrieved subgraph along with the input text sequence, we propose a graph encoding that is permutation and relation inversion invariant yet efficient. Furthermore, to ensure that the model does make use of the encoded knowledge when generating responses, we propose a multi-modal contrastive learning objective between the two different graph-text modalities to enforce the consistency across the retrieved facts and the generated texts. We refer to our framework as **SU**bgraph **R**etrieval-augmented **GE**neration (**SURGE**).

We validate our SURGE framework on the OpendialKG [29] dataset against relevant baselines, with the T5-small [34] model as the base PLM. When evaluating the generated responses from the dialogue agents, the conventional metrics (e.g. BLEU [31], Rouge [27]) can not measure how faithfully the generated responses reflect the world knowledge. Thus, we introduce an additional performance metric, referred to as Knowledge-verifying Question Answering (KQA), which evaluates whether generated responses contain the correct knowledge with an extractive question answering model. The experimental results show that SURGE generates responses that not only agree with the gold knowledge but are also consistent with the retrieved knowledge from the KG.

Our main contributions can be summarized as follows:

• We propose a context-relevant subgraph retrieval method for knowledge graph-augmented dialogue generation, to extract only the relevant piece of the knowledge for the given context from the entire knowledge graph, for generating more appropriate responses to the ongoing conversation.

• We propose an invariant yet efficient graph encoder and a multi-modal graph-text contrastive learning objective to ensure that the generated responses faithfully reflect the retrieved knowledge.

• We validate SURGE against relevant baselines, demonstrating its efficacy in generating responses that are more informative by retrieving and reflecting the relevant knowledge from the KG.

---

[1]In this work, we denote the knowledge as facts (i.e., a set of triplets) in the knowledge graph.

## 2  Related Work

**Pre-trained Language Model**  Large Pre-trained Language Models (PLMs) [32, 23, 34] that use the encoder-decoder architecture based on Transformers [42] have achieved great successes on language generation tasks. As they can accurately contextualize the given context and then generate human-like sentences, they are often used as the base architecture for the neural dialogue systems [49, 15]. Moreover, when the PLMs become larger, dialogue generation models have shown to generate high-quality responses [1], suggesting that pre-trained parameters do contain certain knowledge. However, despite the fluency of such PLM-based dialogue agents, they often generate factually incorrect responses that are unfaithful to the dialogue context but look plausible – widely known as a hallucination problem [28]. Thus, generating responses requiring specific and valid factual knowledge is still challenging. To tackle this, recent works propose to retrieve knowledge from external sources, and then use it to augment the neural dialogue agents [37, 40], discussed below.

**Knowledge-Grounded Dialogue**  The sources of external knowledge can be categorized into two types: documents from large unstructured corpora such as Wikipedia [7] or Web [30], and symbolic facts from Knowledge Graphs (KGs) [2, 43]. Firstly, Dinan et al. [7] propose a retrieval-based dialogue generation model, which links the pre-compiled documents retrieved from Wikipedia articles with the given dialogue context using the information retrieval [3]. Further, several works [19, 25, 40] propose to learn the document retrievers in an end-to-end fashion, to generate the knowledge-grounded responses for the given dialogue. However, KG-augmented dialogue generation models, which use structured KGs, are more efficient than the previous methods utilizing unstructured texts thanks to the efficacy of KG for encoding knowledge, and consequently thus more preferable when responsiveness is important [26]. Regarding the dialogue generation with the KG, Moon et al. [29] introduce a knowledge-grounded dialogue dataset where each dialogue comes with the large-scale KG. Before the era of pre-trained language models, several works [41, 45, 48, 5, 50] have suggested sequence-to-sequence models that generate dialogue by conditioning the output word distribution with the entities from the KG. Further, Galetzka et al. [9] propose an efficient way to encode all of the facts in the 1-hop neighbors of the entities that appear in the dialogue history in the given KG, in order to reduce the number of input tokens used in the pre-trained language model [32]. However, all of these methods simply match and retrieve all the facts for entities including irrelevant ones that appear in the dialogue history, which may mislead the agent into generating out-of-context responses. Our work differs from these existing works, since we aim at retrieving only the context-relevant subgraph among the 1-hop facts with a novel subgraph retriever, which is end-to-end trainable along with the dialogue generation model.

## 3  Method

We first discuss the basic ingredients: graph neural networks and transformers. We then formalize the dialogue generation problem and describe the key components for our **SU**bgraph **R**etrieval-augmented **GE**neration (**SURGE**) framework: context-relevant subgraph retrieval, invariant graph encoding, and graph-text contrastive learning. Figure 2 illustrates the overview of our framework.

### 3.1  Preliminaries

As we use two different modalities, namely text and graph, we first define them, and then describe the neural networks to encode them. In particular, a text is defined as a sequence of tokens $\boldsymbol{x} = [x_1, ..., x_N], \forall x_i \in \mathcal{V}$, where $x_i$ is a token and $\mathcal{V}$ is a pre-defined vocabulary formed with specific tokenization algorithms [39]. On the other hand, a knowledge graph (KG) is a type of multi-relational graphs $\mathcal{G} = \{(\mathsf{e}_h, \mathsf{r}, \mathsf{e}_t)\} \in \mathcal{E} \times \mathcal{R} \times \mathcal{E}$, where $\mathsf{e}_h$ and $\mathsf{e}_t$ are head and tail entities along with their relation $\mathsf{r}$, and $\mathcal{E}$ and $\mathcal{R}$ are sets of entities and relations, respectively, i.e., $\mathsf{e}_h, \mathsf{e}_t \in \mathcal{E}$ and $\mathsf{r} \in \mathcal{R}$.

To easily access different modalities in the same framework, we define the mapping function that maps entities and relations in the KG to the tokens in the text as follows: $q_e : \mathcal{E} \rightarrow \mathcal{V}^l$ and $q_r : \mathcal{R} \rightarrow \mathcal{V}^l$. In other words, any entity $\mathsf{e} \in \mathcal{E}$ and relation $\mathsf{r} \in \mathcal{R}$ can be mapped to a sequence of $l$ tokens $\boldsymbol{x} \in \mathcal{V}^l$: $q_e(\mathsf{e}) = \boldsymbol{x}_e$ and $q_r(\mathsf{r}) = \boldsymbol{x}_r$. Such functions enable us to associate the KG symbol with the text.

**Transformer**  A Transformer [42] is a neural architecture that embeds a sequence of tokens while taking their relationships into account. It is a basic building block of recent PLMs [6, 32]. Formally, assume that we have a sequence of tokens $\boldsymbol{x} = [x_1, ..., x_N], \forall x_i \in \mathcal{V}$, then a goal of generative transformers is to generate a sequence of tokens $\boldsymbol{y}_{<t} = [y_1, ..., y_{t-1}], \forall y_i \in \mathcal{V}$, with encoder Enc, decoder Dec and tokens' embedding function $f$. Thus, a hidden state at time $t$ for generating

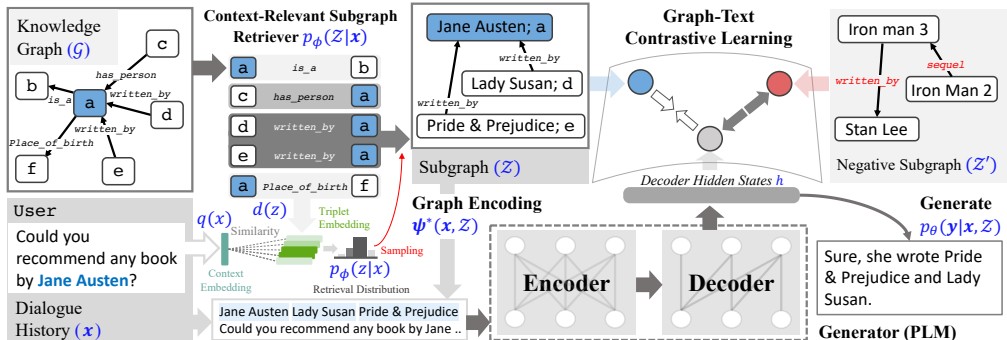

Figure 2: **Framework Overview.** Our framework, SURGE, consists of three parts. First, a context-relevant subgraph retriever $p_\phi(\mathcal{Z}|\boldsymbol{x})$ retrieves the subgraph $\mathcal{Z}$ relevant to the given dialogue history $\boldsymbol{x}$ from a knowledge graph $\mathcal{G}$ (e.g., 1-hop KG from entity *Jane Austen*; a). Specifically, we measure the similarity of a context and triplet embedding to compose the retrieval distribution $p_\phi(z|\boldsymbol{x})$ (§ 3.3). Then, we encode the retrieved subgraph $\mathcal{Z}$ into the input of the generator, using the graph encoding function $\boldsymbol{\psi}(\boldsymbol{x}, \mathcal{Z})$ (§ 3.4). Finally, we use a contrastive learning to enforce the model to generate a consistent response with the retrieved subgraph (§ 3.5).

$y_t$ is $\boldsymbol{h}_t = \texttt{Dec}(\texttt{Enc}(\boldsymbol{X}), \boldsymbol{Y}_{<t})$, where $\boldsymbol{X} = f(\boldsymbol{x}) = [f(x_1), ..., f(x_N)]$ and $\boldsymbol{Y}_{<t} = f(\boldsymbol{y}_{<t}) = [f(y_1), ..., f(y_{t-1})]$. We note that both Enc and Dec functions are **permutation sensitive** with positional encodings as in generic Transformer architecture [42, 46].

**Graph Neural Network** A Graph Neural Network (GNN) represents a node with its neighboring nodes over the graph structure [12], which is formalized as follows:

$$\boldsymbol{e}_t^{(k+1)} = \texttt{GNN}^{(k)}(\boldsymbol{e}_t^{(k)}; \mathcal{G}) = \texttt{UPD}^{(k)}(\boldsymbol{e}_t^{(k)}, \texttt{AGG}^{(k)}(\{\boldsymbol{e}_h^{(k)} \mid \forall \texttt{e}_h \in \mathcal{N}(\texttt{e}_t; \mathcal{G})\})), \tag{1}$$

where $\boldsymbol{e}_t$ and $\boldsymbol{e}_h$ are embeddings of entities (nodes) $\texttt{e}_t$ and $\texttt{e}_h$, respectively, $\mathcal{N}(\texttt{e}_t; \mathcal{G}) = \{\texttt{e}_h \mid (\texttt{e}_h, \texttt{r}, \texttt{e}_t) \in \mathcal{G}\}$ is a set of neighboring entities of $\texttt{e}_t$, AGG is a function that aggregates embeddings of $\texttt{e}_t$'s neighboring entities, and UPD is a function that updates a representation of $\boldsymbol{e}_t$ with the aggregated messages from AGG, at each iteration $k$.

## 3.2 Problem Statement

Here we formalize the problem of context-relevant subgraph retrieval for knowledge-grounded dialogue generation. Given a dialogue history $\boldsymbol{x} = [x_1, \ldots, x_N]$, a model with generative PLMs first encodes the input tokens, and then models a probabilistic distribution $p(\boldsymbol{y}|\boldsymbol{x})$ to generate an output response $\boldsymbol{y} = [y_1, \ldots, y_T]$. This problem requires a piece of specific knowledge for a conversation.

To that end, given a dialogue history $\boldsymbol{x}$, we aim at retrieving a subgraph $\mathcal{Z} \subseteq \mathcal{G}$ consisting of a set of triplets $z \in \mathcal{Z}$ where $z = (\texttt{e}_h, \texttt{r}, \texttt{e}_t)$, which encodes relevant knowledge for ongoing conversation. Thus, the distribution of the context-relevant facts $\mathcal{Z}$ is $p(\mathcal{Z}|\boldsymbol{x})$, and our final likelihood of generating responses then becomes $p(\boldsymbol{y}|\boldsymbol{x}, \mathcal{Z})$. Then, to jointly optimize the objective of graph retrieval with response generation, we treat $\mathcal{Z}$ as a latent variable and then marginalize the likelihood of the generative model over all possible latent variables for retrieved subgraphs $\mathcal{Z}$, formalized as follows:

$$p(\boldsymbol{y}|\boldsymbol{x}) = \sum_{\mathcal{Z} \subseteq \mathcal{G}} p_\phi(\mathcal{Z}|\boldsymbol{x}) \, p_\theta(\boldsymbol{y}|\boldsymbol{x}, \mathcal{Z}) = \sum_{\mathcal{Z} \subseteq \mathcal{G}} p_\phi(\mathcal{Z}|\boldsymbol{x}) \prod_t^T p_\theta(y_t|\boldsymbol{x}, \mathcal{Z}, \boldsymbol{y}_{1:t-1}), \tag{2}$$

where $p_\phi(\mathcal{Z}|\boldsymbol{x})$ is an output distribution of the context-relevant subgraph retriever, and $p_\theta(\boldsymbol{y}|\boldsymbol{x}, \mathcal{Z})$ is the target distribution of a knowledge-augmented generator, parameterized as $\phi$ and $\theta$, respectively.

## 3.3 Context-Relevant Subgraph Retriever

We now provide a concrete description of our context-relevant subgraph retriever formalized in Eq. 2. We assume that a retrieval probability of each triplet in $\mathcal{Z} = \{z_1, \ldots, z_n\}$ is independent. Then, for simplicity, we decompose the probability of retrieving a set of triplets $p(\mathcal{Z}|\boldsymbol{x})$ into the product of individual triplet retrieval probabilities, as follows: $p(\mathcal{Z}|\boldsymbol{x}) = p(z_1|\boldsymbol{x})p(z_2|\boldsymbol{x}) \ldots p(z_n|\boldsymbol{x})$.

From the aforementioned Eq. about $p(\mathcal{Z}|\boldsymbol{x})$, we can now focus on retrieving the only one triplet. Therefore, we define the retrieval of one triplet with an inner product of dense vectors between the dialogue history $\boldsymbol{x}$ and the candidate triplet $z$, similarly to a dense retrieval model [11], as follows:

$$p_\phi(z|\boldsymbol{x}) \propto \exp(d(z)^\top q(\boldsymbol{x})), \tag{3}$$

where $d$ is a triplet embedding function and $q$ is a dialogue context embedding function. We can use a PLM for implementing $q$, but we need another effective method for $d$ that can reflect the property of the graph. Therefore, we propose the GNN-based triplet embedding method for realizing $d$.

Let consider a set of triplets associated to the entities that appear in the given dialogue context $\{(\mathtt{e}, \mathtt{r}, \mathtt{e}_t) \text{ or } (\mathtt{e}_h, \mathtt{r}, \mathtt{e}) \mid q_e(\mathtt{e}) \subseteq \boldsymbol{x}\}$, as the retrieval candidates. To effectively represent the triplets consisting of entities and their relations as items, we use GNNs described in Section 3.1 for the triplet embedding function $d$. In our triplet retrieval, representing both nodes and edges, which are equally essential components for the multi-relational graph, is worthwhile to represent an entire triplet. To do so, we adopt the existing edge message passing framework [17] that transforms edges of the original graph to nodes of the dual hypergraph [38] (i.e., transforming $\mathcal{G}$ to $\mathcal{G}^*$), which allows us to use existing node-level GNNs for representing edges of the original graph (See Section D.1 of the Supplementary File for more details). Formally, our triplet embedding function is denoted as follows:

$$d(z) = \mathtt{MLP}([\boldsymbol{e}_h \parallel \boldsymbol{r} \parallel \boldsymbol{e}_t]), \; \boldsymbol{e}_h = \mathtt{GNN}(\boldsymbol{e}_h; \mathcal{G}), \; \boldsymbol{r} = \mathtt{GNN}(\boldsymbol{r}; \mathcal{G}^*), \; \boldsymbol{e}_t = \mathtt{GNN}(\boldsymbol{e}_t; \mathcal{G}), \qquad (4)$$

where $z = (e_h, r, e_t)$, and $\parallel$ is the concatenation operator.

## 3.4 Invariant Graph Encoding

In this subsection, we then now specify the remaining operations for $p_\theta(\boldsymbol{y}|\boldsymbol{x}, \mathcal{Z})$, which generates $\boldsymbol{y}$ conditioned on the two different modalities, namely text $\boldsymbol{x}$ and graph $\mathcal{Z}$. Before doing so, we first define the notion of graph encoding, whose goal is to leverage the retrieved subgraph information along with the dialogue history for response generation, which is formalized in Definition 3.1.

**Definition 3.1. (Graph Encoding)** *Let $\boldsymbol{\psi}(\boldsymbol{x}, \mathcal{Z})$ be a graph encoding function. Then, given a sequence of tokens $\boldsymbol{x} = [x_1, ..., x_N]$ and a subgraph $\mathcal{Z}$, it first yields a new sequence $\boldsymbol{x}' = [x'_1, ..., x'_m, x_1, ..., x_N]$ where $[x'_1, ..., x'_m]$ comes from $q_e(\mathtt{e}) = x'_e$ and $q_r(\boldsymbol{r}) = x'_r$ $\forall (\mathtt{e}, \boldsymbol{r}, *) \in \mathcal{Z}$. Then, it embeds a sequence $\boldsymbol{X}' = [f(x'_1), ..., f(x'_m), f(x_1), ..., f(x_N)] = f([x'_1, ..., x'_m, x_1, ..., x_N])$, where $f$ is the token embedding function. Consequently, $\boldsymbol{X}' = \boldsymbol{\psi}(\boldsymbol{x}, \mathcal{Z})$.*

For instance, given a sequence $\boldsymbol{x} = [x_1, \ldots, x_N]$ and a subgraph $\mathcal{Z} = \{(\mathtt{a}, \mathtt{d}, \mathtt{b}), (\mathtt{b}, \mathtt{e}, \mathtt{a}), (\mathtt{a}, \mathtt{d}, \mathtt{c})\}$ from the retriever, $\boldsymbol{\psi}(\boldsymbol{x}, \mathcal{Z}) = f([a, d, b, b, e, a, a, d, c, x_1, ..., x_N])$ with $a = q_e(\mathtt{a})$, $b = q_e(\mathtt{b})$, $c = q_e(\mathtt{c})$, $d = q_r(\mathtt{d})$, $e = q_r(\mathtt{e})$, which we term as the naïve encoding. Due to its simplicity, it is widely used for a text-conditioned generation [24]. However, for graph encoding, it violates two important invariance properties: permutation invariance [47] and relation-inversion invariance, which are formalized in Definition 3.2, 3.3.

**Definition 3.2. (Permutation Invariance)** *For any set permutation $\pi$, $\boldsymbol{\psi}(\boldsymbol{x}, \mathcal{Z}) = \boldsymbol{\psi}(\boldsymbol{x}, \pi \cdot \mathcal{Z})$, i.e., an order of elements in a subgraph does not affect a representation.*

**Definition 3.3. (Relation Inversion Invariance)** *Let a relation $\neg \mathtt{d}$ be an inverse relation to $\mathtt{d}$, if $(\mathtt{a}, \mathtt{d}, \mathtt{b}) = (\mathtt{b}, \neg \mathtt{d}, \mathtt{a}) \, \forall \mathtt{a}, \mathtt{b} \in \mathcal{E}$. Then, $\boldsymbol{\psi}(\boldsymbol{x}, \mathcal{Z} \cup \{(\mathtt{a}, \mathtt{d}, \mathtt{b})\}) = \boldsymbol{\psi}(\boldsymbol{x}, \mathcal{Z} \cup \{(\mathtt{b}, \neg \mathtt{d}, \mathtt{a})\})$ for any subgraph $\mathcal{Z}$.*

**Invariant Graph Encoding** To meet both properties, we consider two additional operations on a set of triplets up to the naïve encoding. We first define a SORT operator that returns the same output regardless of the order of input set elements, as follows:

$$\mathtt{SORT}(\pi \cdot \mathcal{Z}) = \mathtt{SORT}(\pi' \cdot \mathcal{Z}), \; \forall \pi, \pi' \in S_n, \qquad (5)$$

where $S_n$ is a set of all possible permutations for $n$ elements. Moreover, we define a INV operator that adds the inverse triplet of each triplet in the subgraph $\mathcal{Z}$, as follows:

$$\mathtt{INV}(\mathcal{Z}) = \mathcal{Z} \cup \{(\mathtt{e}_t, \neg \mathtt{r}, \mathtt{e}_h) \mid (\mathtt{e}_h, \mathtt{r}, \mathtt{e}_t) \in \mathcal{Z}\}. \qquad (6)$$

With above operations, we now define a more solid graph encoding function: $\boldsymbol{\psi}(\boldsymbol{x}, \mathtt{SORT}(\mathtt{INV}(\mathcal{Z})))$, which satisfies both permutation and relation inversion invariance.

**Invariant and Efficient Graph Encoding** However, above encoding is not efficient since it requires the $\mathcal{O}(n)$ space complexity for encoding a graph with $n$ triplets. To be more efficient, we newly define $\tilde{\boldsymbol{\psi}}$ that only encodes the unique nodes (entities) along the sequence, formalized as follows:

$$\tilde{\boldsymbol{\psi}}(\boldsymbol{x}, \mathtt{SORT}(\mathtt{ENT}(\mathcal{Z}))) = f([a, b, c, x_1, \ldots, x_N]),$$

where $\mathtt{ENT}(\mathcal{Z})$ returns the set of unique nodes in $\mathcal{Z}$ and SORT is used to preserve the permutation invariance. This encoding is thus invariant but efficient since it only costs $\mathcal{O}(k)$, for a $k$-entity

sequence where $k < n$. However, as it does not consider the relational information in $\mathcal{Z}$, we further perturb the entities' token embeddings with respect to their representations in $\mathcal{Z}$. Specifically, for each entity $\mathtt{a} \in \mathtt{ENT}(\mathcal{Z})$, we apply affine transformations from learnable Multi-Layer Perceptrons (MLP) on the token embedding of $\mathtt{a}$ as follows:

$$\boldsymbol{\beta}(f(a), \mathcal{Z}) = (1 + \boldsymbol{\gamma}) * f(a) + \boldsymbol{\delta}, \tag{7}$$
$$\boldsymbol{\gamma} = \mathtt{MLP}_1(\boldsymbol{\eta}), \quad \boldsymbol{\delta} = \mathtt{MLP}_2(\boldsymbol{\eta}), \quad \boldsymbol{\eta} = \mathtt{UPD}(f(a), \mathtt{AGGR}(\{f(b), \mathtt{r} \mid \forall \mathtt{b} \in \mathcal{N}(\mathtt{a}; \mathcal{Z})\})),$$

where $\boldsymbol{\beta} : \mathbb{R}^d \to \mathbb{R}^d$ perturbs the embedding according to $\mathcal{Z}$, $\mathtt{AGGR}$ is the relation-aware aggregation function for triplet $(\mathtt{b}, \mathtt{r}, \mathtt{a}) \in \mathcal{Z}$ with $a = q_e(\mathtt{a})$ and $b = q_e(\mathtt{b})$. In sum, we denote a relation-aware invariant and efficient encoder $\boldsymbol{\psi}^*$, formally represented as follows:

$$\boldsymbol{\psi}^*(\boldsymbol{x}, \mathcal{Z}) = \boldsymbol{\beta}(\tilde{\psi}(\boldsymbol{x}, \mathtt{SORT}(\mathtt{ENT}(\mathcal{Z}))), \mathtt{INV}(\mathcal{Z})),$$

where $\boldsymbol{\beta}$ can be applied to the sequence of representations, $\boldsymbol{\beta} : \mathbb{R}^{n \times d} \to \mathbb{R}^{n \times d}$. We conclude that our graph encoding satisfies both properties. For proofs, please see Section C of the Supplementary File.

### 3.5 Consistent Generation with Graph-Text Contrastive Learning

Although the previous schemes allow retrieving and encoding subgraphs that are relevant to the input dialogue history, the consistent generation with the given subgraph is further required, when generating responses with the factual knowledge. In other words, the model should be able to generate different sequences given different graphs with the same dialogue history.

However, we only access the single ground-truth response regardless of the retrieved knowledge, while the generative model is trained with a teacher forcing. Thus, this setting can give rise to the problem of *exposure bias* [35]: the model is never exposed to other generated tokens during training. To overcome such limitations, we introduce a novel graph-text contrastive learning method motivated by multi-modal contrastive learning [33]. Formally, for a single pair of a graph and text, the contrastive learning objective is defined as follows:

$$\mathcal{L}_{cont} = \frac{1}{2} \log \frac{\exp(\mathtt{sim}(\zeta(\boldsymbol{z}), \xi(\boldsymbol{h}))/\tau)}{\sum_{\boldsymbol{h}'} \exp(\mathtt{sim}(\zeta(\boldsymbol{z}), \xi(\boldsymbol{h}'))/\tau)} + \frac{1}{2} \log \frac{\exp(\mathtt{sim}(\zeta(\boldsymbol{z}), \xi(\boldsymbol{h}))/\tau)}{\sum_{\boldsymbol{z}'} \exp(\mathtt{sim}(\zeta(\boldsymbol{z}'), \xi(\boldsymbol{h}))/\tau)}, \quad (8)$$

where $\boldsymbol{z} = \frac{1}{m} \sum_{i=1}^{m} \boldsymbol{z}_i'$ is the mean of graph representations from $[\boldsymbol{z}_1', \ldots, \boldsymbol{z}_m', \boldsymbol{z}_1, \ldots, \boldsymbol{z}_N] = \mathtt{Enc}(\boldsymbol{\psi}^*(\boldsymbol{x}, \mathcal{Z}))$, $\boldsymbol{h} = \frac{1}{T} \sum_{t=1}^{T} \boldsymbol{h}_t$ is the mean of decoder representations, $\mathtt{sim}$ is the cosine similarity, $\zeta$ and $\xi$ are learnable linear projection layers, and $\tau$ is a learnable temperature parameter. Furthermore, $\sum_{\boldsymbol{h}'}$ and $\sum_{\boldsymbol{z}'}$ indicate the summation over negative samples, which are other texts or graphs within a same mini-batch as in previous contrastive literature [4, 10, 13, 20, 22, 33]. With Eq. 8, the model can embed the correlated pairs closer together in order to generate a consistent response to a given graph, i.e., given a different graph, the model would generate different tokens for the same text.

### 3.6 Training

We train the entire model by maximizing the log-likelihood $\log p(\boldsymbol{y}|\boldsymbol{x})$ defined in Eq. 2 with respect to parameters of the retriever $\phi$ and the generator $\theta$. However, computing the marginal probability over all possible subgraphs $\sum_{\mathcal{Z} \subseteq \mathcal{G}} p_\phi(\mathcal{Z}|\boldsymbol{x}) p_\theta(\boldsymbol{y}|\boldsymbol{x}, \mathcal{Z})$ is impractical. Therefore, as in existing works [11, 24], we approximate this by instead summing over $k$ sampled subgraphs. Moreover, for each subgraph, we samples $n$ triplets without replacement from the categorical distribution, parameterized by $\phi$: $z_1, \ldots, z_n \sim Cat(|\mathcal{G}|, p_\phi(z|\boldsymbol{x}))$, which results in $\mathcal{Z} = \{z_1, \ldots, z_n\}$.

Our whole end-to-end training objective for retrieval-augmented generation is then defined as follows:

$$\mathcal{L}_{ret} = \log \sum_{i=1}^{k} p_\phi(\mathcal{Z}_i|\boldsymbol{x}) p_\theta(\boldsymbol{y}|\boldsymbol{x}, \mathcal{Z}_i), \quad \mathcal{Z}_i \sim p_\phi(\mathcal{Z}|\boldsymbol{x}), \tag{9}$$

where we simplify the sampling over $n$ triplets as the sampling over the subgraph distribution $p_\phi(\mathcal{Z}|\boldsymbol{x})$. We assume that we can access the gold subgraph for some data in training. Thus, we further add the supervised retrieval loss to introduce a semi-supervised retriever learning as follows:

$$\mathcal{L}_{sup} = \log p_\phi(\mathcal{Z}^*|\boldsymbol{x}), \tag{10}$$

where $\mathcal{Z}^*$ is the available ground-truth subgraph. Combining all objectives in Eq. 8, 9, and 10, our final training objective is then defined as follows: $\mathcal{L} = \mathcal{L}_{ret} + \mathcal{L}_{sup} + \mathcal{L}_{cont}$.

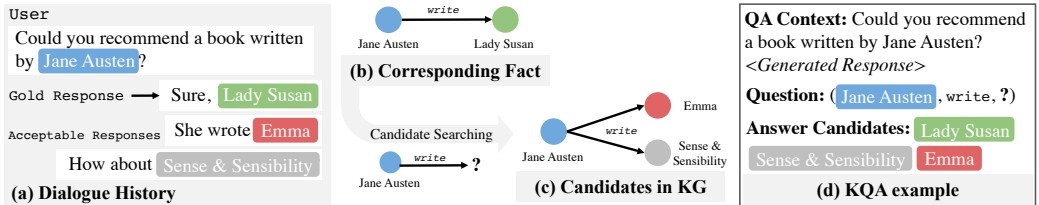

Figure 3: **KQA.** (Left) An example where multiple responses are acceptable but the gold response cannot reflect all of them. (Middle) We first find the fact from the KG that reflects the relation between entities within the user input and gold response (b), and then search candidate facts from the KG (c). (Right) Corresponding KQA example. If a generated response contains the one of answer candidates, the KQA can predict it (success).

## 4 A Novel Metric: Knowledge-verifying QA

Existing automatic evaluation metrics, namely BLEU and ROUGE [31, 27], are limited in that they only consider the lexical overlaps of words without measuring the factual correctness of the generated responses. As shown in Figure 3 (a), there could be multiple correct responses, but existing metrics score them lower due to the lexical mismatch. To solve this issue, we propose **K**nowledge-verifying **Q**uestion **A**nswering (**KQA**) which measures whether generated responses contain factually correct knowledge given the dialogue history. Compared to the existing metrics using question generation methods [14, 44], we automatically derive QA pairs for evaluation from the dialogue and the large-scale KG [2]. In particular, at first, the ground truth is an entity mentioned in the gold response, and a question is composed of an entity in a dialogue and the existing relation between two entities. Once the question is determined, more answer candidates can be found through the searching in KG, which allows more acceptable responses that contain the correct knowledge as in Figure 3 (d).

## 5 Experiment

### 5.1 Experimental Setup

We conduct experiments on the **OpendialKG** dataset [29], which is a dialogue corpus associated with a large-scale Knowledge Graph (KG), namely Freebase [2] with 100k entities and 1M facts. This dataset has 15K dialogues with 91K utterances. We note that $49\%$ of the responses come with the gold knowledge, whereas others are not. Since the dataset does not provide a predefined data split, we randomly split it into train (70%), validation (15%), and test sets (15%). We use **T5-small** [34] for all experiments. We also conduct experiments on another dataset and language model and present the results in Section E of the Supplementary File. For details, see Section D of the Supplementary File.

### 5.2 Baselines and Our Models

We compare different variants of our SURGE framework against various KG-augmented dialogue generation models. **No Knowledge.** This model is only provided with the dialog history, thus no external knowledge is used. **All Knowledge.** This model is provided with entire facts within a 1-hop subgraph of entities associated with the dialog history. **Gold Knowledge.** This model is provided with the exact gold knowledge, even in the test time if the gold knowledge exists. **Space Efficient Encoding.** This model takes all facts from the 1-hop subgraph of the entities as input. We use two different encoding methods introduced in [9], namely Space Efficient (series) and Space Efficient (parallel). **EARL.** This is an RNN-based model, where the entities are conditioned in response generation [50]. **Random/Sparse Retrieval.** These models are provided with selected facts from a 1-hop subgraph, via the random sampling or the sparse retrieval – BM25 [36]. **Text-based Retrieval.** This model is a variant of our framework where T5 encoder [34] is used for $d$ in Eq. 4 instead of GNNs similar to [16]. **SURGE (unsupervised).** Ours with retrieved context-relevant facts from 1-hop subgraph, where the retrieval is trained without any supervision. **SURGE (semi-supervised).** Ours but the retriever is trained with supervision if the data has a gold fact. **SURGE (contrastive).** Our full model jointly trains the retriever in a semi-supervised manner with the contrastive learning term. By default, all our models are trained with an invariant and efficient graph encoding.

### 5.3 Evaluation Metrics

We evaluate the generated responses using BLEU [31], ROUGE [27] and unigram overlap (F1) with the gold response. Along with these conventional text evaluation metrics, we also evaluate the results

Table 1: Experimental results on OpendialKG dataset. † indicates the model under the oracle setting using the gold facts even in the test time.

| | Method | KQA | | BLEU | | | | ROUGE | | | Unigram |
|---|---|---|---|---|---|---|---|---|---|---|---|
| | | EM | F1 | B-1 | B-2 | B-3 | B-4 | R-1 | R-2 | R-L | F1 |
| *Baselines* | **No Knowledge** | 7.62 | 13.2 | 15.79 | 9.19 | 5.61 | 3.43 | 19.67 | 7.13 | 19.02 | 22.21 |
| | **All Knowledge** | 30.06 | 34.95 | 15.95 | 9.98 | 6.72 | 4.65 | 20.96 | 8.50 | 20.21 | 24.34 |
| | **Space Efficient** *(series)* | 26.88 | 31.15 | 16.15 | 10.03 | 6.66 | 4.50 | 21.15 | 8.56 | 20.44 | 24.55 |
| | **Space Efficient** *(parallel)* | 28.90 | 33.19 | 16.33 | 10.22 | 6.81 | 4.64 | 21.42 | 8.85 | 20.68 | 24.87 |
| | **EARL** | 24.52 | 27.09 | 11.49 | 6.34 | 4.06 | 2.75 | 15.36 | 4.37 | 14.61 | 16.88 |
| *Retrieval variants* | **Random Retrieval** | 21.05 | 26.09 | 15.70 | 9.52 | 6.12 | 3.99 | 20.21 | 7.88 | 19.55 | 23.28 |
| | **Sparse Retrieval** (BM25) | 19.32 | 24.55 | 15.63 | 9.44 | 6.05 | 3.96 | 20.05 | 7.67 | 19.37 | 23.10 |
| | **Text-based Retrieval** | 31.00 | 35.95 | 16.87 | 10.64 | 7.23 | 5.07 | 20.63 | 8.53 | 19.89 | 24.16 |
| *Ours* | **SURGE** *(unsupervised)* | 37.35 | 42.24 | 18.10 | 11.65 | 7.99 | 5.59 | 22.14 | 9.50 | 21.23 | 25.91 |
| | **SURGE** *(semi-supervised)* | **39.57** | **44.13** | **18.21** | **11.74** | **8.08** | **5.68** | 22.11 | 9.41 | 21.22 | 25.91 |
| | **SURGE** *(contrastive)* | 39.52 | 43.96 | 17.72 | 11.53 | 7.96 | 5.61 | **22.19** | **9.77** | **21.34** | **25.94** |
| *Oracle* | **Gold Knowledge**† | 49.76 | 53.41 | 18.47 | 12.79 | 9.32 | 6.92 | 24.93 | 11.97 | 24.03 | 28.82 |
| | **Gold Response** | 83.88 | 86.22 | 100.0 | 100.0 | 100.0 | 100.0 | 100.0 | 100.0 | 100.0 | 100.0 |

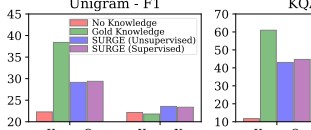

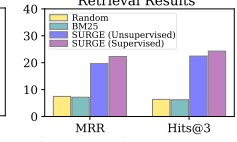

Figure 4: Results of whether gold knowledge exists (Know O) or not (Know X) for the dialogue history. We note that T5 + Gold Knowledge exactly uses the gold knowledge for generating responses.

Figure 5: Knowledge retrieval results on the OpendialKG dataset, with MRR and Hits@3.

Table 2: Results on knowledge-consistent response generation, where we compare three variants of our SURGE – unsupervised, semi-supervised and contrastive, on F1 and KF1 as metrics.

| Method | F1 | KF1 |
|---|---|---|
| **SURGE** (unsupervised) | 27.78 | 24.09 |
| **SURGE** (semi-supervised) | **28.30** | 26.38 |
| **SURGE** (contrastive) | 28.17 | **27.58** |

using our new metric, KQA (Section 4), which measures whether the generated responses contain proper knowledge. Lastly, we compute the Knowledge F1 (KF1) similarly as in Shuster et al. [40], which is a measure of unigram overlap between the retrieved knowledge and generated response.

## 5.4 Experimental Results and Analysis

In Table 1, we report the knowledge-grounded response generation performances of baselines and our SURGE. As shown in Table 1, our models significantly outperform all the baseline models, excluding the upper bound measure, *Gold Knowledge*, in all evaluation metrics. The high BLEU, ROUGE, and F1 refer that ours sufficiently learns the syntactic and semantic structure of the responses. Our models also achieve high F1 and EM scores in KQA. The high KQA scores indicate that the generated responses are formed with the correct facts, which are relevant to the dialog context. Even the baseline models such as *All Knowledge*, *Space Efficient Encoding* [9], and *EARL* [50], which are provided with all of 1-hop facts, underperform than ours. The result demonstrates that selecting relevant knowledge is critical in knowledge-augmented response generation.

Figure 4 examines the generation performance further by categorizing the dialogues into two groups: those with gold knowledge and those without. When there is no gold knowledge, the model using *Gold Knowledge* suffers a significant drop in all metrics and performs similarly to No Knowledge. On the contrary, ours significantly improves the unigram F1 and KQA scores even with the retrieved knowledge without using the exact gold knowledge.

**Knowledge Retrieval**   Figure 5 shows the performances of knowledge retrieval methods, where we only measure the retrieval performance on dialogues that contain the gold knowledge. We use Mean Reciprocal Rank (MRR) and Hits@k as metrics. Note that our SURGE is a differentiable retriever, which jointly learns to retrieve the context-relevant knowledge and then generate the corresponding responses, whereas *Random* and *BM25* [36] retrieve the knowledge without learning. Therefore, our models outperform other retrieval approaches by a large margin (See Section G with Figure 4 of the Supplementary File for examples of baselines and ours). In addition, when our retriever is trained in a semi-supervised manner, we observe the substantial performance gains from unsupervised learning, as the model can learn to retrieve the ground truth knowledge during training.

**Knowledge-Consistent Generation**   We conduct an ablation study on our models to validate the knowledge consistency performance of the response generation by computing the Knowledge F1 (KF1) score [40]. To focus solely on the case where a given knowledge is consistently reflected in the generated responses, we use the gold knowledge rather than the retrieved one. We randomly perturb

| | Context | Gold response | Baseline response | SURGE response |
|---|---|---|---|---|
| (a) | I loved Moby Dick. Can you recommend something similar? | It was written by Herman Melville in 1851. It's sometimes called The Whale. | Moby Dick is a sailor. Do you like her work? | Moby Dick was written by Herman Melville. He also wrote The Whale. |
| (b) | Do you know anything the actor Adam Brown? | Yes, he was in the movie The Hobbit: An Unexpected Journey. | Adam Brown starred in King Kong. Have you seen it? | Adam Brown starred in The Hobbit: The Desolation of Smaug and The Hobbit: The Battle of the Five Armies. |

| (a) Retrieved Subgraph from SURGE | (b) Retrieved Subgraph from SURGE |
|---|---|
| (Moby Dick; or, The Whale, written_by, Herman Melville) | (The Hobbit: The Battle of the Five Armies, starred_actors, Adam Brown) |
| (Moby Dick, written_by, Norman Corwin) | (The Hobbit: An Unexpected Journey, starred_actors, Adam Brown) |
| (Moby Dick, written_by, Ray Bradbury) | (The Hobbit: The Desolation of Smaug, starred_actors, Adam Brown) |

Figure 6: Examples of the baseline (Space Efficient, parallel) responses and SURGE responses. On both examples, the baseline generates statements which are factually wrong while SURGE successfully retrieves appropriate facts and generate the good response.

Table 3: Performance comparisons of variants of graph encodings, described in Section 3.4.

| | KQA | | Knowledge Length |
|---|---|---|---|
| Method | EM | F1 | |
| Naïve | 38.18 | 42.18 | 62 |
| Invariant | 39.54 | 43.28 | 117 |
| **Efficient** (entity only) | 38.80 | 43.06 | 39 |
| **Invariant & Efficient** | **39.57** | **44.13** | **39** |

Table 4: Human evaluation on **Consis**tency, **Info**rmativeness, and **Fluency**.

| Method | Consis. | Info. | Fluency |
|---|---|---|---|
| **All Knowledge** | **2.52** | 1.99 | 2.62 |
| **Space Efficient** | 2.47 | 1.75 | 2.46 |
| **SURGE** (ours) | **2.71** | **2.39** | **2.92** |

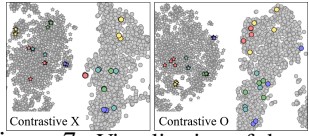

Figure 7: Visualization of the embedding space from our graph(star)-text(circle) contrastive learning.

the gold knowledge to ensure that responses are generated from the given knowledge rather than the trained knowledge. Table 2 shows that our model with a contrastive learning term outperforms all others in the KF1, implying that the generated responses accurately reflect the encoded knowledge.

**Sensitive Analysis on Graph Encoding**    We further conduct an analysis on graph encoding variants introduced in Section 3.4. The knowledge length in Table 3 indicates the average token length used for graph encoding. Our *Invariant & Efficient $\psi^*$* performs the best against other variants, while using the lesser space at the graph encoding phase. Notably, simple *Invariant* achieves a comparable performance against *Invariant & Efficient*, but yields a longer sequence.

**Retrieval and Generation Examples**    Figure 6 shows the examples of generated responses along with the retrieved knowledge. In particular, we compare our SURGE against *Space Efficient (parallel)* baseline. In example (a), the baseline response contains an incorrect fact distracted by the contextually irrelevant entity 'sailor'. Contrarily, SURGE successfully retrieves relevant facts from the KG then generates the factually correct response. This tendency is similar in example (b), where the baseline incorrectly generates the response with a wrong fact containing 'King Kong', meanwhile our SURGE retrieves context-relevant facts and generates a informative response.

**Human Evaluation**    We sample 30 responses of SURGE, *All Knowledge*, and *Space Efficient* on the OpendialKG test dataset [29], then conduct a human study of them. We recruit 46 annotators, and ask them to evaluate the quality of the generated responses by the 3 models given in a random order, with 3 criteria – consistency, informativeness, and fluency – using a 3 point Likert-like scale. As shown in Table 4, ours obtains significantly (p-value $< 0.05$) higher scores than others in all criteria, which is another evidence that our framework generates consistent, informative, and fluent responses.

**Embedding Space Visualization**    We further visualize the multi-modal graph-text latent space in Figure 7. The visualization shows that, for the same dialogue with different subgraphs, our SURGE with graph-text contrastive learning (right) generates distinct response embeddings pertraining to different subgraphs, unlike the one without graph-text contrastive learning which shows less variety over responses for the same dialogue (left). We include zoomed Figure 7 in the **Supplementary File**.

## 6    Conclusion

We proposed a novel end-to-end framework for knowledge graph-augmented dialogue generation which retrieves context-relevant subgraph, encodes a subgraph with the text, and generates knowledge-consistent responses, called as **SU**bgraph **R**etrieval-augmented **GE**neration (**SURGE**). Our results demonstrate the effectiveness of our framework in both quantitative and qualitative experiments in knowledge retrieval and response generation tasks. The analysis shows the contribution of each proposed component: retrieval, encoding, and graph-text representation learning. Our work suggests a new direction to generate informative responses for knowledge graph-based dialogue task by empirically showing the importance of retrieving the more relevant subgraph knowledge rather than using all the relevant knowledge graphs when generating knowledge-grounded responses.

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
