# OpenReview forum: "Knowledge-Consistent Dialogue Generation with Knowledge Graphs"
_NeurIPS.cc/2022/Conference — NeurIPS 2022 Submitted_

### Official Review · Reviewer_JGjW · 2022-07-09

**Rating:** 3
**Confidence:** 4
**Soundness:** 2 fair
**Presentation:** 2 fair
**Contribution:** 2 fair

**Summary:**

This paper proposed a KG-based dialogue generation approach.

The proposed approach consists of a triplet selection, a subgraph construction and retrieval, a graph encoding and a basic encoder-decoder framework.

Contrastive learning is used for ensuring graph-text consistency in generation.

A KQA metric is proposed for the evaluation of generated dialogue in fact-level.

Experiments in a benchmark dataset verified the effectiveness of the proposed approach.

**Questions:**

Please see the weaknesses.

**Ethics Review Area:**

["I don’t know"]

**Limitations:**

The lack of comparisons with SOTA methods limited the novelty of the proposed approach.

The only one experimental dataset limited the extension of the proposed approach and the common conclusions presented in the paper.

**Strengths And Weaknesses:**

Strengths:

1. A novel subgraph construction and retrieval scheme.

2. The proposed invariant graph encoding considered the order and inverse relation of a triplet.

3. The use of a loose evaluation metric for dialogue generation.

Weaknesses:

1. The lack of SOTA methods for empirical comparisons on triplets selection.

2. The lack of verification of different graph-based neural networks for encoding.

3. I am afraid p(Z|x) defined in this paper is not a probability.

4. The lack of empirical comparisons with other triplets ranking methods or using other methods for the performance verification of dialogue generation.

5. A minor point that only one dataset is used in this paper. I concerned the extension of the proposed approach.

6. Another minor point is that there are a lot of grammar errors, a careful proofread is needed.

---

> ### Author Response · Authors · 2022-08-01
> **Initial Response (2/2)**
>
> **Q4.** The lack of empirical comparisons with other triplets ranking methods or using other methods for the performance verification of dialogue generation.
>
> **Answer** Regarding additional empirical comparisons on triplet ranking methods, please refer to the answer for the Q1.
>
> We want to emphasize that we have utilized various automatic metrics (BLEU, ROUGE, F1) and human evaluations to verify the quality of generated responses.
> In addition, we propose the novel metrics named KQA which evaluates the quality of knowledge within the generated responses based on the diverse facts within the large-scale knowledge graph.
>
> But we agree with your point that using more metrics for performance evaluation might highlight the merit of our framework.
> Thus, we additionally measure the Distinct [1] metric to evaluate the diversity of generated responses from baselines and our SURGE framework. Results clearly show that our framework contributes to generating more diverse dialogues than all of the baselines.
>
> |   | Dist-1  | Dist-2  |
> |---|:---:|:---:|
> | No Knowledge  | 6.06 | 15.73  |
> | All Knowledge  | 9.67 | 24.45  |
> | SEE (Series)  | 8.49 | 21.77  |
> | SEE (Parallel)  | 8.78 | 22.70  |
> | EARL  | 5.15 | 16.46  |
> | Sparse Retrieval (BM25)  | 7.65 | 19.63  |
> | Ours (Semi-supervised)  | **10.33** | **28.26**  |
>
> ---
>
> **Q5.** A minor point that only one dataset is used in this paper. I concerned the extension of the proposed approach.
>
> **Answer** We kindly request you to refer to the experimental results on another dataset named KOMODIS in Table 3 of the Supplementary File.
>
> ---
>
> **Q6.** Another minor point is that there are a lot of grammar errors, a careful proofread is needed.
>
> **Answer** Thank you for your suggestion. We will carefully check our paper several times to reduce such grammatical errors.
>
> ---
>
> **References**
>
> [1] Li et al., A Diversity-Promoting Objective Function for Neural Conversation Models, NAACL 2016.

---

> ### Author Response · Authors · 2022-08-01
> **Initial Response (1/2)**
>
> We sincerely thank you for your constructive and helpful comments. We initially address all your concerns below:
>
> ---
>
> **Q1.** The lack of SOTA methods for empirical comparisons on triplets selection.
>
> **Answer** Thank you for your suggestions on more empirical comparisons on triplets selection.
> First of all, we want to emphasize that the sparse retrieval method (BM25) that we already addressed has shown comparable performance against neural baselines on various information retrieval tasks [1].
> However, to faithfully reflect your suggestion, we performed additionally experiments with our SURGE framework with a text-based neural encoder (pre-trained 1 T5 [2] encoder) on the triplet embedding function $d$ instead of the GNN-based encoder, similar to the Bi-encoder setting from the recent work [3].
>
> |   | KQA EM  | KQA F1  | B-1  | B-2  | B-3  | B-4  | R-1  | R-2  | R-L  | F1  |  # parameters for $d$  |
> |---|:---:|:---:|:---:|:---:|:---:|:---:|:---:|:---:|:---:|:---:|:---:|
> | Sparse Retrieval (BM25)  | 19.32 | 24.55  | 15.63  | 9.44  | 6.05  | 3.96  | 20.05  | 7.67  | 19.37  | 23.10  | -  |
> | Text-based Retrieval (T5)  | 31.00  | 35.95  | 16.87  | 10.64  | 7.23  | 5.07  | 20.63  | 8.53  | 19.89  | 24.16  | 30M  |
> | Ours (Semi-supervised)  | **39.57**  | **44.13**  | **18.21**  | **11.74**  | **8.08**  | **5.68**  | **22.19**  | **9.77**  | **21.34**  | **25.94**  | <1M  |
>
> Experimental results show that the GNN-based encoder outperforms the Text-based encoder, showing that the consideration of the graph structure is crucial in retrieving more informative subgraphs for the knowledge-grounded response generation.
> Additionally, the efficiency of our suggested GNN-based triplet embedding function is demonstrated by the fact that the number of GNN parameters is significantly lower than the number of parameters for a pre-trained language model.
>
> We have also included these results in the revision.
>
> ---
>
> **Q2.** The lack of verification of different graph-based neural networks for encoding.
>
> **Answer** Thank you for suggesting the intriguing ablations on GNN for encoding.
> We agree that ablation studies on GNNs used in both triplet embedding function $d$ (Equation 4) and invariant & efficient graph encoding (Equation 7) might provide useful insights for practitioners.
>
> To this end, first of all, we replaced the GCN used in node-level embedding in Equation 4 with the Graph Attention Networks (GAT) [4] to verify the importance of the GNN architecture on the retrieval performance. Results (row 1) show that GAT slightly harms the performance of our framework. This result indicates the use of GCN may suffice to propagate the entity information to its neighbors.
>
> Then, we replace CompGCN used in our graph encoding with GCN to validate that the consideration of the relation information really matters to the generation performance. Results (row 2) show that the use of GCN largely drops the performance since it cannot consider relations between entities, which is an important component to build a relation-aware invariant and efficient encoder $\psi^*$.
>
> |   | KQA EM  | KQA F1  | B-1  | B-2  | B-3  | B-4  | R-1  | R-2  | R-L  | F1  |
> |---|:---:|:---:|:---:|:---:|:---:|:---:|:---:|:---:|:---:|:---:|
> | Ours (Eq4. GCN $\rightarrow$ GAT)  | 38.58 | 43.21  | 18.00  | 11.52  | 7.88  | 5.55  | 21.79  | 9.20  | 20.90  | 25.60 |
> | Ours (Eq7. CompGCN $\rightarrow$ GCN)  | 37.50  | 42.49  | 17.77  | 11.29  | 7.63  | 5.28  | 21.62  | 9.07  | 20.76  | 25.39 |
> | Ours (Semi-supervised)  | **39.57**  | **44.13**  | **18.21**  | **11.74**  | **8.08**  | **5.68**  | **22.19**  | **9.77**  | **21.34**  | **25.94** |
>
> We will include these ablations in the paper to give more intuitive evidence on GNN design choices.
>
> ---
> **Q3.** I am afraid p(Z|x) defined in this paper is not a probability.
>
> **Answer** We carefully checked the part about $p(Z|x)$ based on your review. However, we found that we clearly explain this in lines 156-159, where the probability $p(Z|x)$ is the product of individual independent triplet retrieval probabilities.
> If you find any concerns on this, please let us know in detail so that we can correct the statement if it is incorrect.
>
> ---
>
> **References**
>
> [1] Thakur et al., BEIR: A Heterogeneous Benchmark for Zero-shot Evaluation of Information Retrieval Models, NeurIPS 2021 Track on Datasets and Benchmarks.
>
> [2] Raffel et al., Exploring the Limits of Transfer Learning with a Unified Text-to-Text Transformer, JMLR 2019.
>
> [3] Humeau et al., Poly-encoders: Transformer Architectures and Pre-training Strategies for Fast and Accurate Multi-sentence Scoring. ICLR 2020.
>
> [4] Velickovic et al., Graph Attention Networks, ICLR 2018.

---

> ### Author Response · Authors · 2022-08-08
> **We sincerely thank you for your efforts in reviewing our paper**
>
> Dear Reviewer JGjW,
>
> We sincerely appreciate your time and effort in reviewing our paper. We believe that your comments would strengthen our paper more solidly.
>
> During the author response period, we have made every effort to faithfully address all your comments/concerns regarding the suggested improvement points, in both the response comment and the paper revision. Therefore, we kindly request you to read our response and then reconsider the evaluation of our work.
>
> We thank you again for your time and efforts in reviewing our paper. Please let us know if you have anything else we should address.
>
> Best regards, Authors

---

> ### Author Response · Authors · 2022-08-09
> **The end of the discussion phase is approaching, and we address all your comments**
>
> Dear Reviewer JGjW,
>
> We have made every effort to faithfully address all your comments in the responses. Here, we briefly summarize the main points of our responses below:
>
> - Regarding SOTA methods on triplet selection, we have added the experimental results with a **powerful text-based retrieval**, which utilizes pre-trained LM for the triplet selection, on which we show our SURGE outperforms it. (Q1)
>
> - Regarding variation on different graph neural networks (GNNs), we have additionally performed **extensive sensitivity analyses on GNNs** used in our framework by replacing them with other GNN variants. (Q2)
>
> - Regarding other evaluation metrics, we have additionally provided the evaluation results with the **Distinct** metric. (Q4)
>
> - Regarding other datasets, we already compared various models on **another dataset KOMODIS in our initial manuscript**, which is provided in the supplementary file. (Q5)
>
> For details, could you please see our response? We sincerely appreciate your insightful and constructive comments. Also, we thank you again for your time and efforts in reviewing our paper, and please let us know if you have any further questions.
>
> Best regards, Authors

---

> ### Author Response · Authors · 2022-08-09
> **Gentle Reminder; Looking forward to your feedback**
>
> Dear Reviewer JGjW,
>
> Thank you for your valuable comments and suggestions again. We are looking forward to any further discussions that would help your re-assessment of our work.
>
> Thanks, Authors

---

### Official Review · Reviewer_Pv3X · 2022-07-09

**Rating:** 6
**Confidence:** 4
**Soundness:** 3 good
**Presentation:** 3 good
**Contribution:** 3 good

**Summary:**

- The authors introduce a method, SURGE (subgraph retrieval augmented generation) for generating consistent dialogue responses, augmented with knowledge from knowledge graphs. SURGE consists of several elements. The first is a context-relevant knowledge retriever, which simply retrieves knowledge from a graph given a dialogue context. The second is an invariant graph encoding; given a set of entity-relation tuples from a subgraph, the authors propose a method for jointly encoding the context and knowledge such that the encoding is invariant to both the order in which facts are presented, and the order in which entities are presented (i.e., relation-invariant). Finally, to encourage consistent generation, the authors propose a contrastive learning objective, which encourages model graph encodings (context, knowledge) to be similar to their decoded output representations, and dissimilar to decoded output representations given other sets of knowledge; as the authors say, “given a different graph, the model would generate different tokens fro the same graph” (L235).
- The authors compare their method against others on the OpenDialKG dataset, measuring standard generation metrics as well as a new proposed Knowledge QA metric, in which the authors examine the performance of a QA model when given a question and model response as input. The authors find that their method outperforms baselines handily, in all metrics considered. Human evaluations and several ablation studies support the authors’ claims and methods.


**Questions:**

- Several times, the authors mention that KG models are more efficient or preferred more than those with unstructured knowledge (e.g., L36-37, L100-103); could you please explain why that is the case?
- The authors mention that the Enc and Dec functions from a transformer are “permutation sensitive” (L135); is it not the case that Enc could in theory be permutation insensitive, due to bidirectional attention?
- Do you have some clarity as to which specific situations the contrastive learning is better than using just semi-supervised, given that, in Table 1, both KQA and BLEU performance is better for the latter?


**Limitations:**

The authors adequately address several limitations within the supplementary material.

**Strengths And Weaknesses:**

### Originality
- **Strengths**: The exact methods themselves may not be new (retrieval-augmented semi-supervised learning, contrastive learning, etc.), however the authors indeed provide a novel combination of techniques. The invariant graph encoding is novel as well, and it is clear how the work differs from previous contributions.

### Quality
- **Strengths**: The authors’ claims are generally well supported and technically sound; the authors present a complete work with several contributions that improve significantly compared to baseline performance. Human evaluations emphasize the gap between the authors’ proposed work and previous methods. The choices of using retrieval-training and invariance encodings are well supported with ablation studies.
- **Weaknesses**: While the three proposed main models all outperform baselines (SURGE with unsupervised, semi-supervised, and contrastive learning losses), it is not clear which aspect of the model truly shines as the best contribution. The gap between SURGE and using sparse retrieval seems to imply that graph retrieval is driving nearly all of the performance gains; this intuitively makes sense, given that OpenDialKG was built using the Freebase KG. Beyond that, however, different methods yield different results; i.e., in some cases, semi-supervised outperforms contrastive, and vice-versa. The performance of naive encoding of (context, subgraph) is also not too far behind the invariance proposals.

### Clarity
- **Strengths**: The paper is well written and clearly outlines the proposed methods and the rationale for using them. Results are generally well presented.
- **Weaknesses**: Figures and tables are very small and compact. While this is not as big of an issue when presenting numbers of graphs, this sees to really hurt other visualizations; Figure 2 (the framework overview) is a bit tough to follow, as are the results in Figure 7.

### Significance
- **Strengths**: This paper provides several methods that other researchers could use in their work, e.g., the invariance encodings (which is helpful in situations with lots of knowledge conditioning), or the various ways in which to train the retriever (e.g. via contrastive learning with the decoder representations). The paper also highlights how a good pipeline can improve performance significantly on a dialogue task requiring knowledge from a knowledge graph.
- **Weaknesses**: The contributions may be limited to instances where a knowledge graph is the most appropriate knowledge source; though the authors claim that using such a knowledge source is more efficient than using unstructured documents, it still limits the types and amount of knowledge at one’s disposal, and it is not immediately clear how to apply this method to situations in which unstructured knowledge is preferred.

---

> ### Author Response · Authors · 2022-08-01
> **Initial Response (2/2)**
>
> **Q5.** Question 2. The authors mention that the Enc and Dec functions from a transformer are “permutation sensitive”; is it not the case that Enc could in theory be permutation insensitive, due to bidirectional attention?
>
> **Answer** Enc function is also permutation sensitive if the positional embeddings are given. We will clarify this point in the corresponding section. Thank you for pointing it out.
>
> ---
>
> **Q6.** Question 3. Do you have some clarity as to which specific situations the contrastive learning is better than using just semi-supervised, given that, in Table 1, both KQA and BLEU performances are better for the later?
>
> **Answer** As discussed in the response to **Q1**, contrastive learning enables the model to learn meaningful latent representations of two modalities, graph and text, where the positive instance pairs of graph and text learn a high similarity between them and the other negative instance pairs learn a low similarity. Similarly, CLIP[1] allows the model to learn aligned representations between two modalities, image and text, which has enhanced the adaptiveness and generalization of the model to various multi-modal downstream tasks.
>
> To observe the impact of contrastive learning in our task, we conducted a constrained experiment in which we exposed our models to handle unseen knowledge graphs. This experiment shows the effectiveness of contrastive learning used in our framework, which enables the pre-trained LM to generate more consistent responses with the given knowledge. In Table 2, the high KF1 score implies that the contrastive learning-based model is better in faithfully reflecting unseen knowledge graphs to responses and aligning two different modalities in the same latent space than other models.
>
> Furthermore, we measure the cosine similarity between hidden representations after the projection layer $\zeta, \xi$  from both modalities of positive and negative pairs to quantitatively analyze the multi-modal alignment. The similarity scores shown below demonstrate that **our contrastive learning-based model has properly learned the aligned latent space** between graph and text.
> Note that below examples are sampled from the test set.
>
> **Graph1:**
> (The Dark Knight Rises, directed_by, Christopher Nolan)
> (The Dark Knight Rises, ~sequel, The Dark Knight)
> (Christopher Nolan, ~produced by,The Dark Knight Rises)
> |Generated Response1 | Similarity Score1|
> |:---|---:|
> | Sure! The Dark Knight Rises is a thriller and a Thriller. Do you like thrillers? | 0.95 |
> | Christopher Nolan also wrote Batman Begins. Have you seen that one?| 0.04 |
> | I have not. I love his song The Last Stand. Have you heard it? | 2.1e-7 |
>
> **Graph2:**
> (Batman Begins, directed_by,	Christopher Nolan)
> (The Prestige, directed_by, Christopher Nolan)
> |Generated Response2 | Similarity Score2|
> |:---|---:|
> | Sure! The Dark Knight Rises is a thriller and a Thriller. Do you like thrillers? | 0.002 |
> | Christopher Nolan also wrote Batman Begins. Have you seen that one?| 0.99 |
> | I have not. I love his song The Last Stand. Have you heard it? | 1.6e-9 |
>
> ---
>
> **References**
>
> [1] Radford et al., Learning transferable visual models from natural language supervision, ICML 2021.

---

> ### Author Response · Authors · 2022-08-01
> **Initial Response (1/2)**
>
> We sincerely thank you for your constructive and helpful comments. We initially address all your concerns below:
>
> ---
>
> **Q1.** Quality; It is not clear which aspect of the model truly shines as the best contribution.
>
> **Answer** Thank you for your insightful comment.
> We want to emphasize that all three components contribute to the best performance of our framework. In Table 1, we show all the performances of our models, trained in different manners. Compared to unsupervised and semi-supervised SURGE in Table 1, the performance gain of contrastive learning of semi-supervised may not be apparent since the contrastive learning is designed to learn latent representation between the retrieved knowledge graph and hidden state of the response generator, which enforces the model to reflect the retrieved knowledge graph faithfully in the generated responses.
>
> The performance of the generator of SURGE is highly dependent on the retrieval performance. If the retriever fails to retrieve the relevant knowledge graphs, the “contrastive” model is inevitable in generating incorrect responses, resulting in slightly worse performance than the “semi-supervised” model. We conduct controlled experiments as explained in lines 318-322 to genuinely evaluate the effect of contrastive learning and present results in Table 2. The high KF1 scores of the “contrastive” model indicate that contrastive learning strengthens our model to generate consistent responses that contain entities emerging in the encoded knowledge.
>
> In terms of graph encoding, we have already shown that the use of graph encoding which meets invariance property shows better performance than naive encoding in Table 3.
>
> ---
>
> **Q2.** Significance; The contributions may be limited to instances where a KG is the most appropriate knowledge source.
>
> **Answer** We appreciate you bringing this important point to our attention.
> Since our work focuses on solving the problem of generating the knowledge-consistent dialogue when a **knowledge graph (KG)** is given as the knowledge source, it could be portrayed that our approach is only limited to a KG dataset, which narrows its applicability in unstructured document tasks.
>
> However, we believe that any text can be converted into a triplet, the base unit of a KG. There is an overhead cost of reformatting the text; however, recent work [1] empirically shows the effectiveness and generalization of knowledge-grounded tasks in transforming unstructured documents into unified knowledge representation, i.e. KG.
>
> ---
> **Q3.** Clarity; Figure 2 and Figure 7 are a bit tough to follow.
>
> **Answer**
> Thank you for raising concerns about the clarity. We will add a larger version of Figure 7 in Supplementary Files for better visibility.
>
> ---
>
> **Q4.** Question 1. Several times, the authors mention that KG models are more efficient or preferred more than those with unstructured knowledge; Could you please explain why that is the case?
>
> **Answer** Thanks for the good question.
> Our aim here is to introduce the knowledge graph as a more efficient counterpart of the unstructured document for encoding the knowledge.
>
> For instance, assume the case illustrated in Figure 1 where the assistant responds to the user’s utterance, “Could you recommend any book by Jane Austen?”.
> When the dialogue agent utilizes the unstructured document (e.g., Wikipedia) as the knowledge source, it should first select the context-relevant paragraphs from the Wikipedia page “Jane Austen” [2]. In this case, the agent needs to understand and analyze the natural language to find the proper knowledge for the response. Intuitively, it is highly complicated to process the unstructured text, even though it might contain more abundant knowledge. The trade-off between informativeness and computation cost cannot be neglected for dialogue agents.
>
> On the other hand, when the dialogue agent leverages the KG (e.g., Wikidata, Freebase), the agent can access refined knowledge regarding the topic [3]. The model with KG can infer relevant knowledge more efficiently than the model with unstructured documents because it contains knowledge in the form of a triplet (factual connections between entities). In other words, the agent can easily find the proper knowledge by attending to the relation “notable work”.
>
> Although the KG requires additional costs to collect, there are numerous sources of KGs available today. We want to highlight again that we focus on developing a method that effectively leverages the KG for the generation of knowledge-grounded dialogues in the case where such an efficient knowledge base exists.
>
> ---
> **References**
>
> [1] Li et al., Knowledge-Grounded Dialogue Generation with a Unified Knowledge Representation, NAACL 2022.
>
> [2] https://en.wikipedia.org/wiki/Jane_Austen
>
> [3] https://www.wikidata.org/wiki/Q36322

---

> ### Author Response · Authors · 2022-08-08
> **We sincerely thank you for your efforts in reviewing our paper**
>
> Dear Reviewer Pv3X,
>
> We sincerely appreciate your time and effort in reviewing our paper. We believe that your comments would strengthen our paper more solidly.
>
> During the author response period, we have made every effort to faithfully answer all questions in your comments in the response and address them on the paper revision. Therefore, we kindly request you to read our response and provide us feedback based on the response.
>
> We thank you again for your time and efforts in reviewing our paper. Please let us know if you have any further questions.
>
> Best regards, Authors

---

> ### Comment · Reviewer_Pv3X · 2022-08-08
> **Response to authors**
>
> Thank you for your response. I believe you have adequately addressed my concerns; I am still leaning towards an accept -- and perhaps slightly more than before -- though my score remains the same as before after reading the other reviews. But I do think that you indeed have clarified several contributions.

---

> > ### Author Response · Authors · 2022-08-09
> > **Thank you for your response**
> >
> > Dear Reviewer Pv3X,
> >
> > We sincerely appreciate your positive comments that you believe our paper is on the acceptance side. Note that we did our best to faithfully address all the comments from all the reviewers during the rebuttal period, and we hope our answers clearly address most of the reviewers' concerns/questions sufficiently. Thus, the concerns from the other reviewer's initial reviews might be resolved through our comprehensive revision as well as the response processes, and we kindly request you to reconsider the rating after other reviewers correct their critical misunderstandings of our work by reading our responses.
> >
> > We thank you again for your time and efforts in reviewing our paper.
> >
> > Best regards, Authors

---

### Official Review · Reviewer_75s5 · 2022-07-13

**Rating:** 4
**Confidence:** 4
**Soundness:** 2 fair
**Presentation:** 2 fair
**Contribution:** 2 fair

**Summary:**

The work aims at avoiding generating factually incorrect responses by introducing knowledge from knowledge graph (KG) into a conversation. It retrieval only context-relevant sub-graph from the global KG to avoid unnecessary encoding overhead and propose an invariant graph encoder to encode the graph.

**Questions:**

None

**Limitations:**

limitations are sufficiently considered in Supplementary materials.

**Strengths And Weaknesses:**

Strength:

The proposed new graph encoding achieves permutation invariance and relation inversion invariance, which I like.

Weaknesses:

1. Inadequate baseline. Some classic and recent methods are not compared or discussed. For example, [1][2][3]

2. Missing ablation study. The effectiveness of the newly-proposed graph encoder is unclear as there is no ablation experiment.

[1]Wu, Sixing, et al. "Diverse and informative dialogue generation with context-specific commonsense knowledge awareness." Proceedings of the 58th annual meeting of the association for computational linguistics. 2020.

[2]Zhang, Houyu, et al. "Grounded conversation generation as guided traverses in commonsense knowledge graphs." arXiv preprint arXiv:1911.02707 (2019).

[3]Cui, Fuwei, et al. "Syntactically Diverse Adversarial Network for Knowledge-Grounded Conversation Generation." Findings of the Association for Computational Linguistics: EMNLP 2021. 2021.

---

> ### Author Response · Authors · 2022-08-01
> **Initial Response**
>
> We sincerely thank you for your constructive and helpful comments. We initially address all your concerns below:
>
> ---
> **Q1.** Inadequate baseline. Some classic and recent methods are not compared or discussed.
>
> **Answer**
> Thank you for suggesting these works. We believe that discussing them will help improve the quality of our paper.
> However, we want to emphasize that all mentioned works do not utilize the pre-trained language model, therefore, it is inappropriate to compare our method against them. Furthermore, we have already discussed the most recent and similar work, EARL [1], and compared it against our framework, demonstrating that our framework outperforms the method without the pre-trained language model.
> ---
> **Q2.** Missing ablation study. The effectiveness of the newly-proposed graph encoder is unclear as there is no ablation experiment.
>
> **Answer**
> We kindly remind you that we have already shown the ablation results in Table 3 if you are referring to the invariant and efficient graph encoding in Section 3.4.
>
> If you are referring to the GNN-based triplet embedding function $d$ in Equation 4, we agree with your concern about the ablation study.
> To address your concern, we experiment SURGE framework with a text-based neural encoder (pre-trained T5 [2] Encoder) on the triplet embedding function $d$ instead of the GNN-based encoder, similar to the Bi-encoder setting from the recent work [3], and then provide the results in the table below.
>
> |   | KQA EM  | KQA F1  | B-1  | B-2  | B-3  | B-4  | R-1  | R-2  | R-L  | F1  |  # parameters for $d$  |
> |---|:---:|:---:|:---:|:---:|:---:|:---:|:---:|:---:|:---:|:---:|:---:|
> | Sparse Retrieval (BM25)  | 19.32 | 24.55  | 15.63  | 9.44  | 6.05  | 3.96  | 20.05  | 7.67  | 19.37  | 23.10  | -  |
> | Text-based Retrieval (T5)  | 31.00  | 35.95  | 16.87  | 10.64  | 7.23  | 5.07  | 20.63  | 8.53  | 19.89  | 24.16  | 30M  |
> | Ours (Semi-supervised)  | **39.57**  | **44.13**  | **18.21**  | **11.74**  | **8.08**  | **5.68**  | **22.19**  | **9.77**  | **21.34**  | **25.94**  | <1M  |
>
> The results of the experiments show that the GNN-based encoder outperforms the Text-based encoder even with fewer parameters, demonstrating the importance of considering the graph structure when generating knowledge-based responses. Note that we also included these results in the revised version of the paper.
>
> ---
> **References**
>
> [1] Zhou et al., EARL: Informative Knowledge-Grounded Conversation Generation with Entity-Agnostic Representation Learning, EMNLP 2021.
>
> [2] Raffel et al., Exploring the Limits of Transfer Learning with a Unified Text-to-Text Transformer, JMLR 2019.
>
> [3] Humeau et al., Poly-encoders: Transformer Architectures and Pre-training Strategies for Fast and Accurate Multi-sentence Scoring. ICLR 2020.

---

> ### Author Response · Authors · 2022-08-08
> **We sincerely thank you for your efforts in reviewing our paper**
>
> Dear Reviewer 75s5,
>
> We sincerely appreciate your time and effort in reviewing our paper. We believe that your comments would strengthen our paper more solidly.
>
> During the discussion period, we have made every effort to faithfully address all your comments/concerns regarding baselines and ablation studies, in both the response comments and the revision. Therefore, we kindly request you to read our responses and then reconsider the evaluation of our work.
>
> We thank you again for your time and efforts in reviewing our paper. Please let us know if you have any further questions.
>
> Best regards, Authors

---

> ### Author Response · Authors · 2022-08-09
> **The end of the discussion phase is approaching, and we address all your comments**
>
> Dear Reviewer 75s5,
>
> We have made every effort to faithfully address all your comments in the responses. Here, we briefly summarize the main points of our responses below:
>
> - [Baseline] We have already compared our method against the most recent and similar method, **EARL**, where our SURGE significantly outperforms it. (See Q1)
>
> - [Ablation Study] We have clarified that our graph encoder for triplet retrieval is beneficial, by comparing it against a **powerful text-based retrieval model** that utilizes pre-trained LM for selecting relevant triplets, on which we show that our SURGE outperforms it. (See Q2)
>
> We sincerely appreciate your insightful and constructive comments. We thank you again for your time and efforts in reviewing our paper, and please let us know if you have any further questions.
>
> Best regards, Authors

---

> ### Author Response · Authors · 2022-08-09
> **Gentle Reminder; Looking forward to your feedback**
>
> Dear Reviewer 75s5,
>
> Thank you for your valuable comments and suggestions again. We are looking forward to any further discussions that would help your re-assessment of our work.
>
> Thanks, Authors

---

### Author Response · Authors · 2022-08-01
**General Response**

Thank you for your efforts in reviewing our paper and insightful reviews of our work. We believe that addressing your feedback will help us further improve the quality of our paper. In addition, we appreciate that reviewers find the proposed method strong and novel, and the performance improvement against baselines remarkable.

First of all, we would like to highlight that we have faithfully addressed **all your concerns** raised in the reviews by providing additional experimental results and analysis. To sum up,
* [75s5, JGjW] We provide the additional ablation with a text-based retriever, which utilizes the powerful **pre-trained T5 encoder** [1] instead of the GNN we proposed.
* [JGjW] We provide one more additional evaluation metric (Distinct [2]) to show that our method generates more diverse responses than other baselines.
* [JGjW] We conduct various ablations on GNNs used in both the triplet embedding function in Equation 4 and graph encoding in Equation 7.

We also address all of the comments from reviewers in the main paper and the supplementary file, and the modified content is highlighted in blue text.

Furthermore, we want to emphasize that our paper has significant merits to be published.
* In this paper, we tackle challenging and fundamental issues of knowledge-consistent dialogue generation with a Knowledge Graph. Our work is significant since it focuses on critical aspects of the field of knowledge-grounded dialogue generation, which is becoming increasingly important as the need for utilizing symbolic knowledge sources, as well as previous document-based knowledge sources, grows [3].
 * Our framework has significant novelties in three fundamental aspects:
1) **Context-Relevant Subgraph Retriever** allows the model to retrieve meaningful subgraphs from the KG, considering the graph structure with the graph neural network.
2) **The Invariant and Efficient Graph Encoder** aims to efficiently combine representations from the text and graph modalities into one, where we carefully design the graph encoding function while taking both invariances into account.
3) **Graph-Text Contrastive Learning** focuses on generating the text responses that faithfully reflect the retrieved subgraph, which is a new way of enforcing consistency across the knowledge and the generated response.
* In experiments, we empirically demonstrate that our framework significantly outperforms baselines on KG-based dialogue datasets. We also conduct extensive ablation studies to thoroughly examine the proposed framework and show that all of the novel components contribute to improving the quality of knowledge-consistent dialogue generation.

We kindly request reviewers to revise their reviews and scores if you find that our responses satisfactorily address all of your concerns. If not, please let us know by the response so that we can improve the quality of our work beyond the current state.

---
**References**

[1] Raffel et al., Exploring the Limits of Transfer Learning with a Unified Text-to-Text Transformer, JMLR 2019.

[2] Li et al., A Diversity-Promoting Objective Function for Neural Conversation Models, NAACL 2016.

[3] Li et al., Knowledge-Grounded Dialogue Generation with a Unified Knowledge Representation, NAACL 2022.

---

### Meta-Review · Area_Chair_ufUS · 2022-08-24

**Recommendation:** Reject
**Confidence:** Certain

**Metareview:**

The paper presents a method for dialogue generation with knowledge graphs, where the goal is to increase faithfulness towards the provided knowledge graph. The research topic, while somewhat narrow, is well-motivated, and they propose a context-relevant subgraph retrieval method (with specialized graph encoding method preserving the permutation/relation inversion invariant objective, new loss function for the generation that encourages consistency with the knowledge subgraph), which shows a promising performance on OpendialKG benchmark dataset.

While I do some values and contributions in this paper, I am not convinced this merits a publication at NeurIPS — each component of the pipeline (contrastive learning, etc) is not novel and it addresses a fairly small domain. If they show this can be applied to more diverse settings, for example, different datasets (I saw that results on KOMODIS are presented in supplementary material, but they are not explained carefully and not very convincing as is. I’d recommend integrating it into the main paper). I also suggest novel metric (section 4) should be more carefully verified before being used to evaluate systems — e.g., how they align with human evaluation, etc.

It is unfortunate that two reviewers did not respond to the author's responses, the area chair examined the paper independently (and also read through the review/responses) to write this meta-review.

**Award:**

No

---

### Decision · Program_Chairs · 2022-09-14

Reject